**Data Availability Statement:** The high-throughput sequence data have been deposited in the National Center for Biotechnology Information (NCBI) BioProject database with project number

# Changes of bacterial and fungal communities and relationship between keystone taxon and physicochemical factors during dairy manure ectopic fermentation

Ping Gong[1], Daoyu Gao[1,2], Xiuzhong Hu[1], Junjun Tan[1], Lijun Wu[1], Wu Liu[1], Yu Yang[1]*, Erguang Jin[1]*

1 Institute of Animal Husbandry and Veterinary, Wuhan Academy of Agricultural Science, Wuhan, Hubei, P.R. China, 2 College of Animal Science and Technology, Yangtze University, Jingzhou, China

* yang007yu@outlook.com (YY); jeg0617@126.com (EJ)

## Abstract

### Background

Due to interactions with variety of environmental and physicochemical factors, the composition and diversity of bacteria and fungi in manure ectopic fermentation are constantly changing. The purpose of this study was to investigated bacterial and fungal changes in dairy manure ectopic fermentation, as well as the relationships between keystone species and physicochemical characteristics.

### Methods

Ectopic fermentation was carried out for 93 days using mattress materials, which was combined with rice husk and rice chaff (6:4, v/v), and dairy waste mixed with manure and sewage. Physicochemical characteristics (moisture content, pH, NH4+-N (NN), total organic carbon (TO), total nitrogen (TN) and the C/N ratio) of ectopic fermentation samples were measured, as well as enzymatic activity (cellulose, urease, dehydrogenase and alkaline phosphatase). Furthermore, the bacterial and fungal communities were studied using 16S rRNA and 18S rRNA gene sequencing, as well as network properties and keystone species were analyzed.

### Results

During the ectopic fermentation, the main pathogenic bacteria reduced while fecal coliform increased. The C/N ratio gradually decreased, whereas cellulase and dehydrogenase remained at lower levels beyond day 65, indicating fermentation maturity and stability. During fermentation, the dominant phyla were *Chloroflexi*, *Firmicutes*, *Proteobacteria*, *Bacteroidetes*, and *Actinobacteria* of bacteria, and *Ascomycota* of fungi, while bacterial and fungal community diversity changed dramatically and inversely. The association between physicochemical characteristics and community keystone taxon was examined, and C/N ratio was negative associated to keystone genus.

PRJNA871233 and PRJNA871235. All relevant data are within the paper and its Supporting information files.

**Funding:** This research was funded by the Innovation Projects of Wuhan (CXJC201902). The funders had no role in study design, data collection and analysis, decision to publish, or preparation of the manuscript.

**Competing interests:** The authors have declared that no competing interests exist.

## Conclusion

These data indicated that microbial composition and diversity interacted with fermentation environment and parameters, while regulation of keystone species management of physico-chemical factors might lead to improved maturation rate and quality during dairy manure ectopic fermentation. These findings provide a reference to enhance the quality and efficiency of waste management on dairy farm.

## Introduction

The livestock business in China has grown quickly over the previous few decades, with livestock output estimated to exceed 18,000 million head by 2020 [1, 2]. With the rapid development of the dairy industry, the resultant large amount of dairy manure, was produced approximate $4 \times 10^8$ ton annually according to statistical data from China's Ministry of Agriculture. Untreated and arbitrary waste discharge causes serious pollution in the surrounding environment, as well as severely limiting the dairy industry's sustainable and healthy development [3, 4]. Furthermore, excessive agricultural sewage emissions can lead to groundwater pollution, which had become one of major threat environment and human health [4]. Hence, sustainable and efficient management of livestock manure is quite urgent to facilitate its handling, transportation and application with limited environmental pollution [5, 6].

Ectopic fermentation system (EFS), composed of complex microbial communities mixed with litter, was developed to overcome the problem of pollution from existing farm wastewater systems [7]. It had been evidenced by several studies in effectively managing pollution problems related to livestock wastes such as cow, pig, and so on [4, 8–10]. The EFS is a bulking material-padded experimental apparatus, which is dynamically fermented by inoculation with a composite microbial agent and continual addition of animal wastes [11]. With the features in large-scale disposal of organic solid wastes, EFS has attracted growing attention from applied researchers and engineers as an *ex situ* decomposition technique [10]. EFS was a unique and ecologically acceptable way of treating both excrement and urine, and the discharged waste and post-fermentation padding may be utilized as bio-fertilizers [8].

During the fermentation process, it is vital to verify whether the primary pathogenic microorganism composition meets national organic fertilizer guidelines. Microbial fermentation beds degrade manure, and microorganisms perform the function of material energy transformation, which is the core in benign operation of the fermentation bed [12, 13]. Because the fermentation bed comprises a diverse microbial population, changes in community and function may have vital impact on the process and results [4, 14, 15]. On the other hand, EFS may reduce the primary zoonotic bacteria found in manure, such as *Salmonella spp*, *Campylobacter spp*, *Listeria monocytogenes*, *Yersinia enterocolitica*, *Escherichia coli* and *protozoa* [9]. Yang et al. (2018) found that thermophilic bacteria especially *Bacillus* have an important role in EFS by 16S rRNA gene sequencing. Previous studies suggested that bacteria-fungi interactions with physiochemical variables might have an impact on composting technology. However, rare study revealed dynamic changes and network properties about bacteria and fungi, also the relation among bacteria, fungi and environmental and physicochemical parameters during EFS.

The purpose of this study was to investigate the dynamic changes of the main pathogenic microorganism, physicochemical properties, microbiota and fungi during EFS. Our study aimed to explore the bacterial and fungal diversity and composition, as well as the influence of

environmental and physicochemical characteristic on the variation of bacteria and fungi during EFS of dairy manure with rice husk and rice chaff.

## Materials and methods

### Experimental materials

This experiment was carried out in an EFS with a total volume 1000 m$^3$ (48 m×12 m×1.8 m). To create the EFS bed, rice husk and rice chaff was mixed at a ratio 6:4 (v/v) as mattress materials. Dairy waste mixed by manure and sewage was periodically added onto the bedding material and turned by the upender for fully mixing, loosening and ventilation to provide oxygen for pig manure fermentation and promote the degradation. Ectopic fermentation samples at day 0, 1, 3, 8, 15, 22, 30, 37, 44, 51, 58, 65, 72, 86 and 93 were collected. On each sampling day, six samples were collected randomly from different sites of the fermentation substrate and immediately pooled. Then, the samples were divided into two parts on ice, and transferred to the laboratory. One part was stored at −80˚C for microbial analysis by sequencing. The other part was used for the determination of physical and chemical properties of the EFS.

### Detection of main pathogenic microorganisms in fecal contamination

Sewage samples were collected according to GB/T 25169–2010. The total number of mold, *Coliform*, *salmonella* and *Staphylococcus aureus* were detected by dilution plate counting method with aseptic operation in ultra clean machine [16]. Briefly, 1 ml sewage (or 1 g waste) was draw into 9 ml saline and then sufficient oscillated. Pre-diluting mixed liquid to different gradient ($10^{-2}$, $10^{-3}$, $10^{-4}$, $10^{-5}$, $10^{-6}$, $10^{-7}$), each 1 ml dilution from two or three adjacent samples was transferred to a petri dish with nutrient AGAR medium at 46˚C. Each gradient had two repeats. After 72 h cultivating at 30˚C, total Bacteria were counted. Three samples with adjacent 100 μL dilutions were coated on the EMB plate, SS plate, and B-P plate, respectively. Each gradient was repeated for two times. The samples were cultured at 37˚C for 24h. The number of *Coliform bacteria*, *Salmonella* and *Staphylococcu* were counted. Results were expressed as log value [lg (CFU/ mL)] of microbial quantity in 1g fecal contamination. The value of *fecal coliform* was determined by multiple-tube fermentation [17].

### Analysis of physicochemical parameters

Fresh samples were used for measurement of moisture content, pH, $NH_4^+$-N (NN), total organic carbon (TO), total nitrogen (TN) and the C/N ratio using previously described methods in Chinese National Agricultural Organic Fertilizer Standard (NY525-2012) [18].

### Enzymatic activities measurements

To measure the enzymatic activities, fresh air-dried samples were prepared. Cellulose, urease, dehydrogenase and alkaline phosphatase were determined by Cellulose Assay Kit (BC0125, Solarbio, Beijing, China), Urease (UE) Assay Kit (BC0155, Solarbio, Beijing, China), Dehydrogenase (UE) Assay Kit (BC0395, Solarbio, Beijing, China) and alkaline phosphatase Assay Kit (BC0280, Solarbio, Beijing, China), respectively. All measurements were performed according to the manufacturer's instructions.

### Microbial analysis

**DNA extraction and high- throughput sequencing.** Microbial DNA was extracted from frozen fermentation samples using a QIAamp DNA Stool Mini Kit (Qiagen, Germany)

following the manufacturer's protocols. Successful DNA extraction was confirmed by 0.8% agarose-gel electrophoresis. Bacterial 16S rRNA and fungal 18S rRNA gene was amplified using primers 341F (5′-ACT CCT ACG GGA GGC AGC AG-3′) and 806R (5′-GGA CTACHV GGG TWT CTA AT-3′) and 547F (5′-CCA GCA SCY GCG GTA ATT CC-3′) and V4R (5′-ACT TTC GTT CTT GAT YRA-3′) respectively. The PCR conditions were pre-denaturation at 98°C for 2 min, 25 cycles of denaturation at 98°C for 15 s, annealing at 55°C for 30 s, elongation at 72°C for 30 s, and a final post-elongation cycle at 72°C for 5 min. The PCR products were purified with AMPure XP beads (AXYGEN). After purification, the PCR products were used for the construction of libraries and then paired-end sequenced on Illumina Miseq (Illumina, CA, USA) at the Personalbio, Shanghai, China.

**Sequence filtering, OTU clustering, and sequence analyses.** The double-ended FASTQ sequences were filtered using the sliding window method, and then FLASH (v1.2.7, http://ccb. jhu.edu/software/FLASH/) was used to align the sequences that passed the quality filtering step [19]. The FASTA and QUAL files from Mothur were converted to FASTQ format using USEARCH (Version 7.0) [20]. Downstream processing and operational taxonomic unit (OTU) identification were performed using UPARSE [21]. Barcodes and primers from the merged sequences were removed. After dereplication and abundance sorting were performed, where singletons were retained, sequences with a minimum similarity of 97% were clustered into OTUs using the average neighbour algorithm. Alpha-diversity analyses, including community richness index (Chao1), community diversity index (Shannon), and community evenness (Pielou_e) determinations were performed using Mothur [22].

**Network analysis.** Microbiota sequencing data from EFS samples to perform network analysis. Absolute abundance data were used for correlation analysis in R to determine correlation between every genus and every other genus within each individual microbiota sample. To avoid the bias introduced by different microbial OTU numbers, we selected the relative abundances of the 100 most abundant genus for microbial and fungal group. Robust correlations between two genera were defined as those with SparCC correlation coefficients > 0.85 and false discovery rate-corrected P-values < 0.01 for bacterial community and correlation coefficients > 0.6 and false discovery rate-corrected P-values < 0.05 for fungal community. The topology parameters including degree, centrality, clustering coefficient and average shortest path length of each network were determined in Cytoscape 3.8.0 using Network Analyzer [23].

## Statistical analysis

Data on pathogenic microorganisms, physiochemical properties, alpha diversity were assessed by ANOVA. The procedure for the repeated measurements of SAS (SAS Institute, Inc., Cary, NC, United States) was used. Data were given as means ± standard errors of the means. A mean comparison was performed using the Duncan's Multiple Range test method with a significant level of $P < 0.05$. We used a nonparametric Mann–Whitney test to determine the variance of topology parameters between microbial and fungal network. In the figures, $P < 0.05$ indicates statistical significance ($^*P < 0.05$, $^{**}P < 0.01$, $^{***}P < 0.001$). Correlations were analyzed by using SparCC and correlation coefficients $>0.5$ and false discovery rate-corrected $P$-values $< 0.05$ were defined robust correlations between two variates. Analyses were performed using R (R Core Team, Vienna, Austria), Graphpad Prism (version 8.0.1, Graphpad Software Inc, La Jolla, California, USA) and SAS (version 9.4; SAS Inst. Inc., Cary, NC).

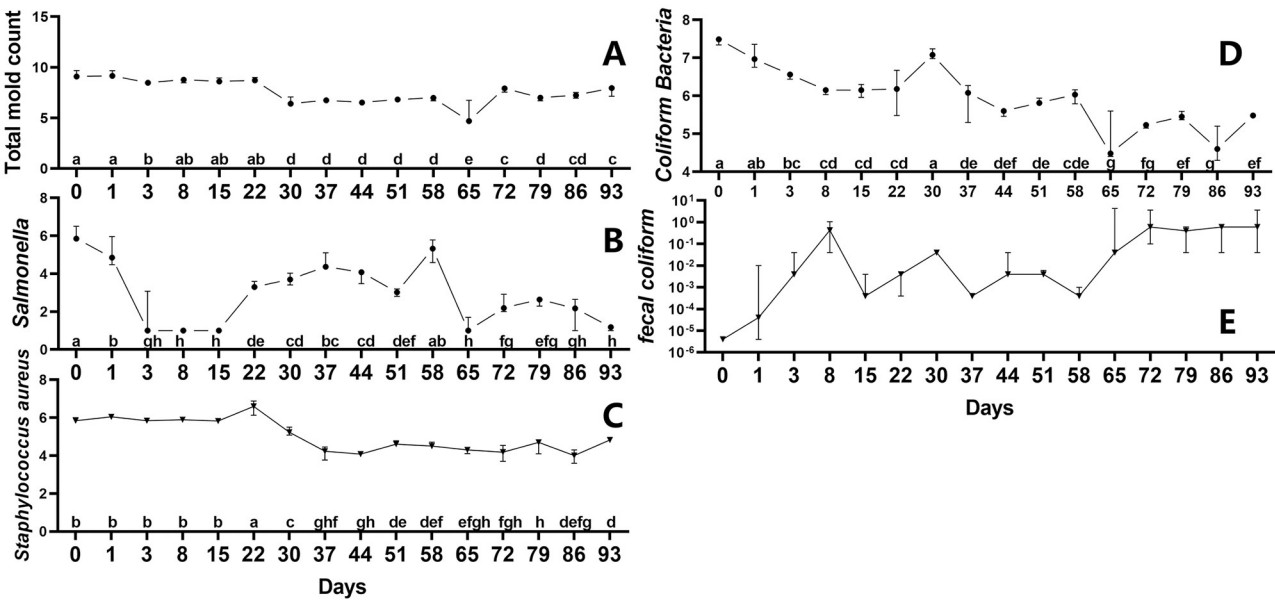

**Fig 1. Changes of pathogenic bacteria and fecal coliform in ectopic fermentation bed.** Count of total mold (A), *Salmonella* (B), *Staphylococcus aureus* (C), *Coliform Bacteria* (D) and *fecal coliform* (E) were assessed during the EFS process. Statistical assessment was carried out with one-way ANOVA followed by post hoc test using Dunn's multiple comparison test. Values with different letters are significantly different at $P < 0.05$.

## Results

### Changes of main pathogenic microorganisms during the ectopic fermentation of dairy manure

The dynamic changes of total mold count (A), *Salmonella* (B), *Staphylococcus* aureus (C), *Coliform Bacteria* (D) and *fecal coliform* (E) during ectopic fermentation process were shown in Fig 1. The count of total mold (Fig 1A) at the early stage (day 0 to 22) of the ectopic fermentation was significantly higher than that of other time period ($P < 0.01$). Afterwards, it was decreased from day 30 to 65, but subsequently increased from day 72 to 93. During the fermentation process, the number of *Salmonella* changed substantially, decreasing significantly from day 0 to 8, increasing again from day 8 to 37, and then decreasing significantly at day 65 (Fig 1B). It gradually decreased in the last 20 days and was lower than on the first day (Fig 1B). *Staphylococcus aureus* count dropped after the fermentation process, with peaked at day 22 and steadied from day 37 to 93 (Fig 1C). Although the amount of *Coliform Bacteria* of at day 30 was significantly higher than that of other periods ($P < 0.01$), it gradually decreased over the fermentation process (Fig 1D). *Fecal coliform* is the minimum sample size for detecting a fecal coliform unit, and is an important hygienic index for systematically evaluating the harmless effect of manure treatment [24]. In fermentation process, the *fecal coliform* value reached a peak value at day 8, and increased significantly from day 65 to 93, reaching the greatest value at day 65, 72 and 93 (Fig 1E). Thus, ectopic fermentation might minimize the danger of fecal coliform in dairy manure.

### Changes of physicochemical characteristics during fermentation

The fermentation process was indicated by measuring the physicochemical properties of pH, TO, TN, C/N, moisture, and NN (Fig 2). The TO fluctuated over the fermentation period, beginning at a low level and remaining at a rather high level from day 8 to 65 before declining

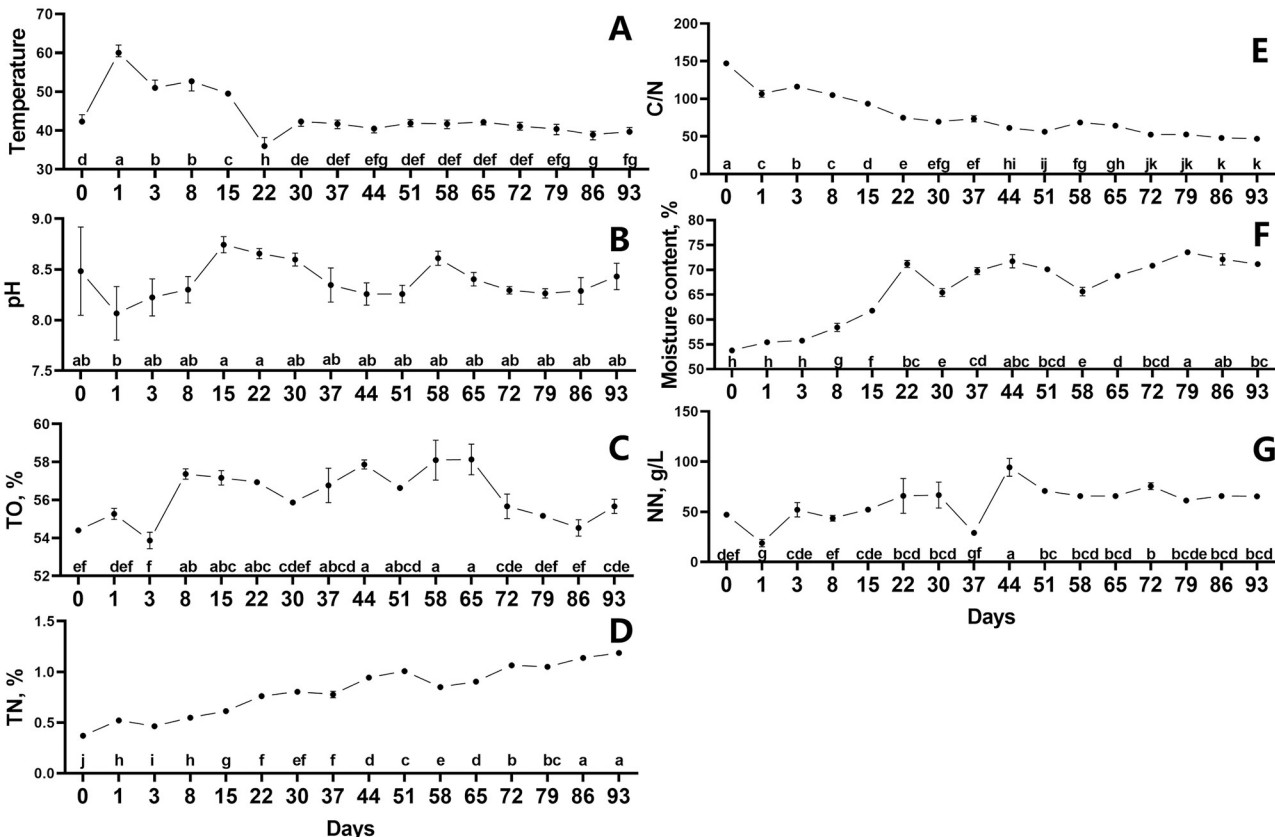

**Fig 2. Changes of physiochemical parameters in ectopic fermentation bed.** Temperature (A), pH (B), TO (C), TN (D), C/N (E), moisture content (F) and NN (G) were assessed during the EFS process. Statistical assessment was carried out with one-way ANOVA followed by post hoc test using Dunn's multiple comparison test. Values with different letters are significantly different at $P < 0.05$.

at the end (Fig 2B). Furthermore, TN increased significantly throughout the fermentation period, whereas C/N decreased gradually (Fig 2C and 2D). The pH was originally reduced on the first day of fermentation, then steadily raised until day 15, then dropped again until day 51. As the fermentation continued, the pH decreased slightly from day 58 to 79, then increased from day 79 to 93 (Fig 2E). The NN was around 50 g/L at start, and there were two drops at day 1 and 37 respectively, which was gradually increased from day 1 to 30 and decreased from day 44 to 93 (Fig 2F). Moisture of the fermentation bed gradually increased in first 22 days followed a fluctuation during day 22 to 58, after which it increased marginally and ultimately reached 71.14% (Fig 2G).

## Changes of enzymatic activities during fermentation

To better comprehend potential pathways of organic matter breakdown, enzymatic activity of cellulose, urease, dehydrogenase, and alkaline phosphatase were examined (Fig 3). Cellulase activity gradually increased over the first 15 days, then fell on day 22, with a minor rise from day 30 to day 44 (Fig 3A). During the fermentation, urease fluctuated and showed three obvious peak value at day 1, day 22 and day 51, respectively (Fig 3B). Dehydrogenase activity climbed quickly to the peak value from day 0 to day 18 and then fell during the next 14 days, while it decreased from day 44 to day 93 (Fig 3C). From day 0 to day 22, alkaline phosphatase

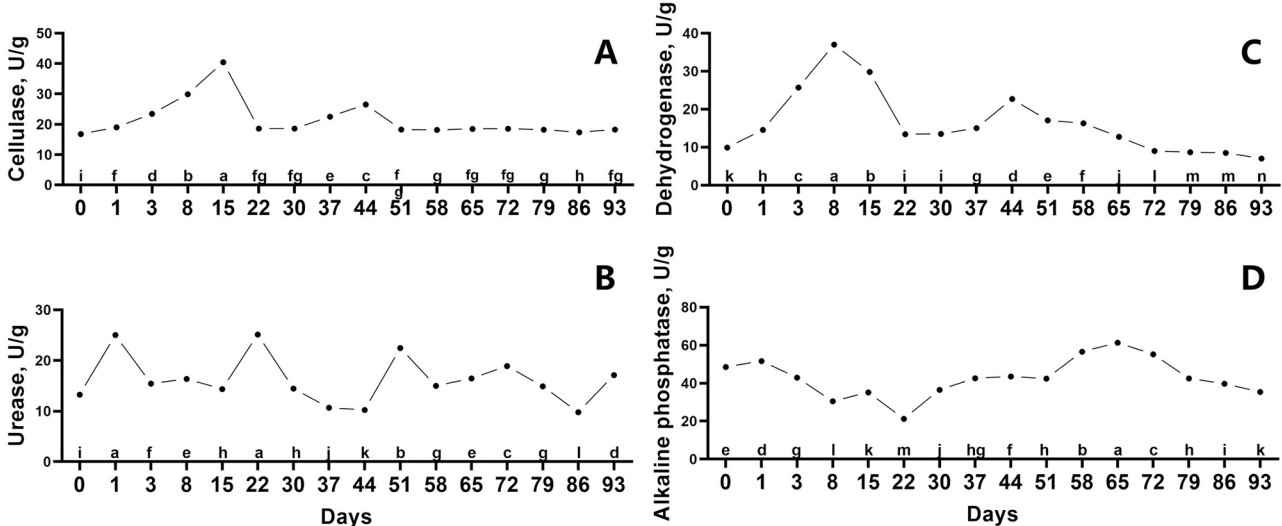

**Fig 3. Changes of enzymatic activities in ectopic fermentation bed.** Cellulase (A), urease (B), dehydrogenase (C) and alkaline phosphatase (D) were determined during the EFS process. Statistical assessment was carried out with one-way ANOVA followed by post hoc test using Dunn's multiple comparison test. Values with different letters are significantly different at $P < 0.05$.

activity progressively declined (Fig 3D). Then, it increased over the following 43 days before declining once more (Fig 3D).

## Analysis of bacterial diversity in ectopic fermentation of dairy manure

We evaluated the diversity changes during the EFS to demonstrate the overall shift of the microbial community (Fig 4). The Chao1 index increased dramatically between days 15 and 86 after being stable for the first eight days, until it started to decline somewhat on day 93 (Fig 4A). Meanwhile, the Shannon index indicated a substantial reduction from day 0 to 15, followed by a rise from day 15 to 86, with two drops at days 58 and 72 (Fig 4B). Pielou_e index is a common species evenness index based on the evenness of the distribution of importance between specie [25]. Pielou_e index likewise displayed a clear decline throughout the first 15 days, followed by an uptick from day 15 to day 22, a plateau from day 22 to day 51, and an uptick from day 58 to day 86 with a decline occurring on day 72 (Fig 4C). However, Bray-Curtis distance analysis of diversity in PCoA revealed a significant change from day 0 to day 22, especially between days 3 and 15 (Fig 4D). There were only minor fluctuations in variety from day 22 to day 93 (Fig 4D).

## Changes of bacteria in ectopic fermentation of dairy manure

Top five bacterial phyla were *Chloroflexi*, *Firmicutes*, *Proteobacteria*, *Bacteroidetes*, and *Actinobacteria*, which were accounted for more than 90% of all bacteria (Fig 5A). *Firmicutes*, *Proteobacteria* and *Bacteroidetes* were prevalent in the microbiota community at beginning of EFS but they reduced afterwards (Fig 5A). Meanwhile *Actinobacteria* was marginally elevated upon EFS (Fig 5A). *Chloroflexi* dramatically inceased and became dominant after EFS (Fig 5A). The relative abundance of *Chloroflexi* peaked on day 15 of ectopic fermentation and subsequently rapidly declined (Fig 5A). Relative abundance of *Firmicutes* peaked on day 8 and subsequently rapidly declined (Fig 5A). The relative abundance of *Proteobacteria* progressively declined (Fig 5A). The relative abundance of *Bacteroidetes* declined progressively, peaked at day 15 of

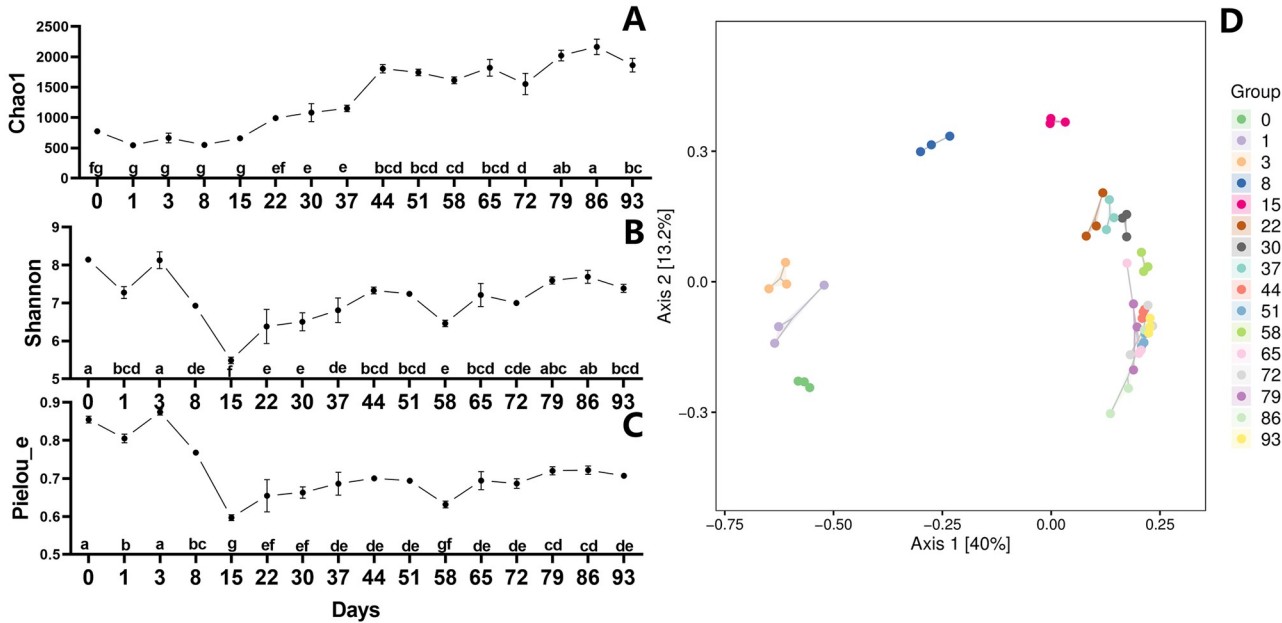

**Fig 4. Changes in bacterial alpha and beta diversity during fermentation.** Chao1 (A), Shannon (B), Pielou_e (C), and Principal coordinate analysis (PCoA) (D) were assessed during the EFS process. Statistical assessment was carried out with one-way ANOVA followed by post hoc test using Dunn's multiple comparison test. Values with different letters are significantly different at $P < 0.05$.

fermentation, and then gradually rose (Fig 5A). As shown in Fig 5B and 5C, *Streptococcus*, *Prevotella*, *Lactobacillus*, *Bifidobacterium*, *Acinetobacter* and *Comamonas* was dominant at day 0. Besides, *Clostridium*, *Ureibacillus*, *Coprococcus*, *Pseudoxanthomonas* and *Thermovum* was enriched in day 1 to 8 (Fig 5B and 5C). The *Sobibacillus* was enriched in day 22, while *B-42*, *Hydrogenophaga* and *Methylocaldum* was enriched in day 65 to 79 (Fig 5B and 5C). At day 65 to 93, *vadinCA02*, *Thauera*, *Ruminofilibacter*, *Sedimentibacter* and *Syntrophomonas* was enriched (Fig 5B and 5C). Additionally, the predominant genera at the end of fermentation were *Clostridium*, *Thauera*, *Hydrogenophaga*, *Sedimentibacter* and *B-42* (Fig 5B and 5C).

## Changes of fungal diversity in ectopic fermentation of dairy manure

The Chao1, Shannon, and Pielou_e indexes were used to assess the development of fungal diversity change (Fig 6). The Chao1 index maintained a plateau from day 0 to day 8, then fell quickly from day 8 to day 22 before becoming almost constant after that (Fig 6A). The Shannon and Pielou_e indexes, meanwhile, followed a similar path to the Chao1 index, remaining neutral for the first eight days, then declining noticeably after a week before changing to a minor change (Fig 6B and 6C). Principal Component Analysis (PCoA), which was used to quantify the diversity, showed that there was a substantial difference between days 3 and 22 even though the first three days and subsequent time periods were the same (Fig 6D).

## Changes of fungi in ectopic fermentation of dairy manure

*Ascomycetes*, *Mucoromycota* and *Basidiomycota* were the most prevalent fungal phyla at day 0 and accounted for more than 70% of all fungi (Fig 7A). From day 0 to 8, *Mucoromycota* decreased from 30% to less than 5%, and *Ascomycetes* showed slightly change (Fig 7A). *Basidiomycota* increased throughout the course of the first eight days, but then began to decline, going from 3.22 percent on day 8 to nearly nonexistent by day 22 (Fig 7A). *Chytridiomycota*

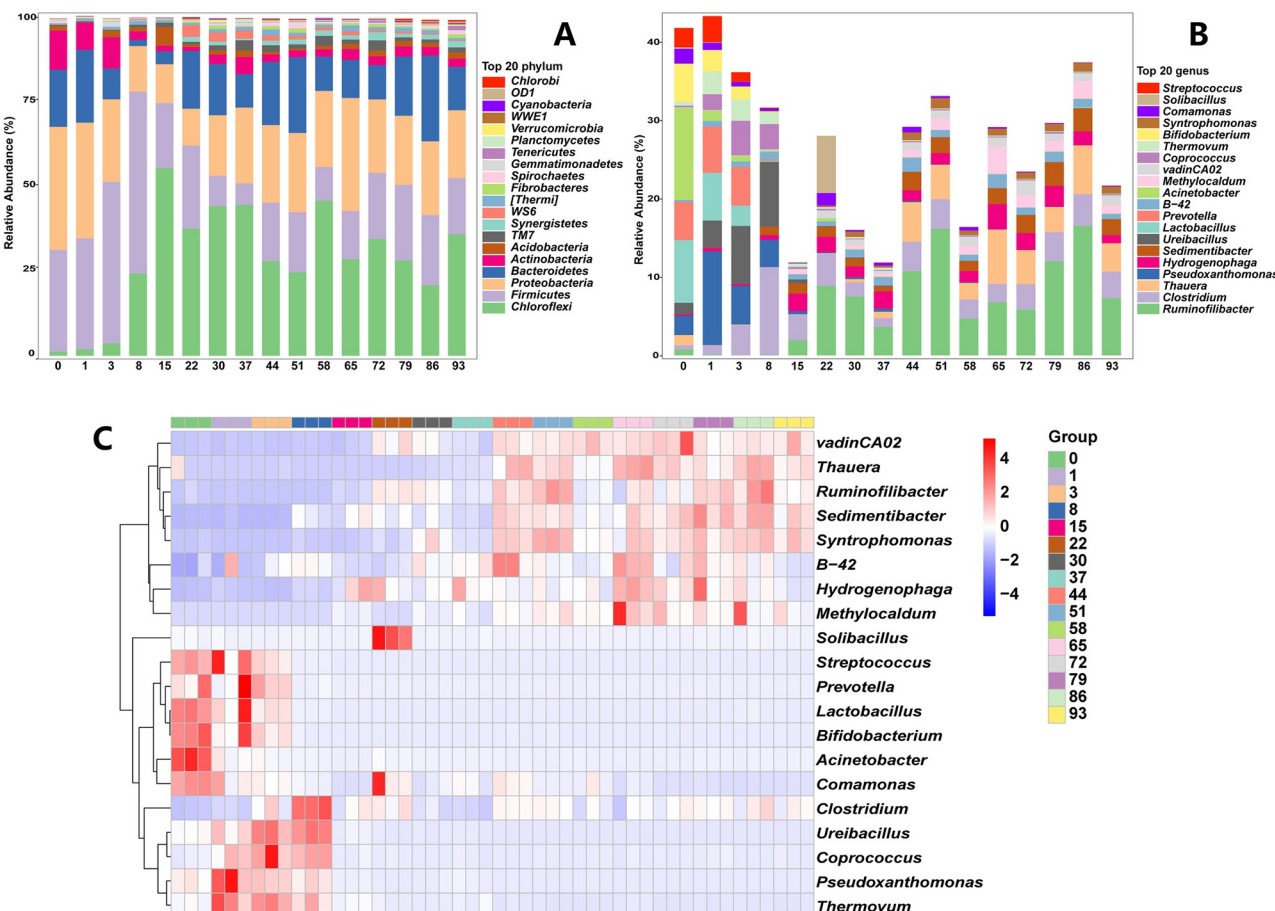

**Fig 5. Changes in bacterial phylum and genus level and heatmap of top 20 genus in sample in ectopic fermentation of dairy manure.**

was increased from day 22 to 51, which eventually disappeared (Fig 7A). At genus level, *Aspergillus*, *Rhizormucor* and *Thermomyces* were dominant at day 0, and subsequently declined through day 15 (Fig 7B). Meanwhile, *Mycothermus*, which was likewise prominent in the identified species, started to rise fast, reaching its greatest level at day 15, and then abruptly decreased on day 22 (Fig 7B). The heatmap also displayed genus-specific alterations that occurred mostly between days 0 and 8 of EFS (Fig 7C). Day 0 showed greater enrichment in *Aspergillus*, *Rhizomucor*, *Thermomyces*, *Tilletia*, *Moeszipmyces*, and *Ustulagininoidea* (Fig 7C). At day 1, *Issachenkia*, *Cephalotrichum* and *Fusarium* were slightly increased (Fig 7C). From day 3 to 8, the abundant genus turned to *Issachenkia*, *Pseudallescheria*, *Sarocladium*, *Lecanicillium* and *Simplicillium* (Fig 7C). All other genera showed moderate abundances in the EFS after day 8, and only *Mycothermus* remained the leading genus with the bulk of readings remaining unidentified.

## Network analysis of microbial communities and correlation between keystone genus and environmental factors

The co-occurrence network was analyzed by SparCC for bacteria and fungi respectively. Bacterial communities demonstrated a more intricate network than fungus, with 25 nodes and 57 edges as opposed to fungi's 10 nodes and 15 edges (Fig 8A and 8B). The bacterial network

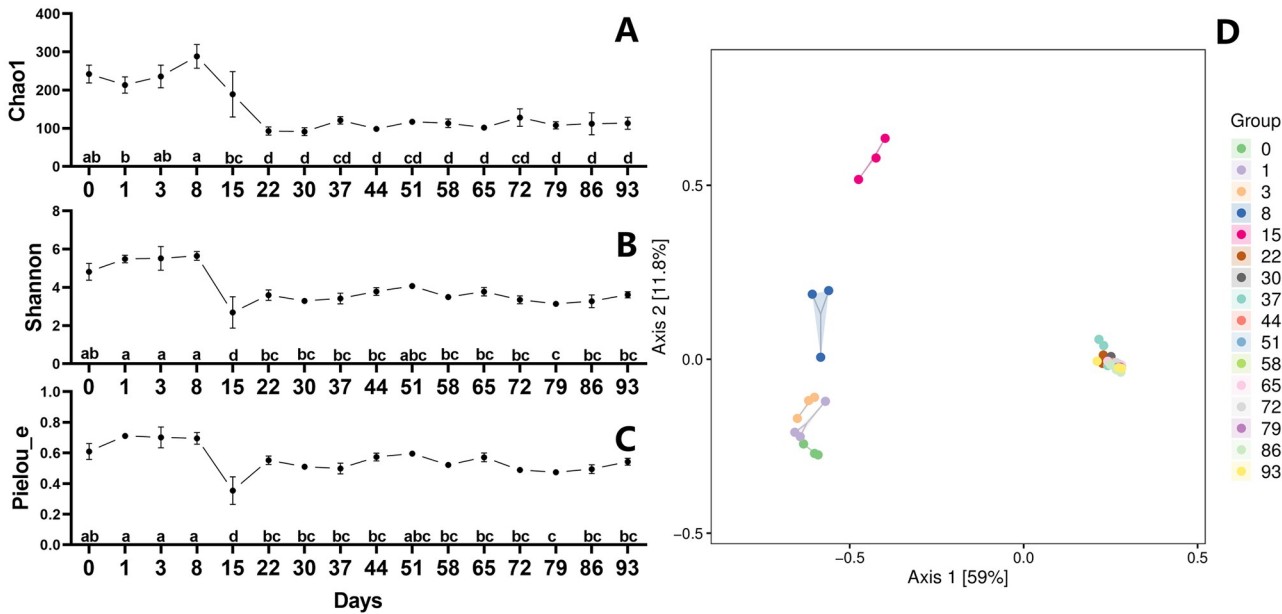

**Fig 6. Changes in fungal alpha and beta diversity during fermentation.** Chao1 (A), Shannon (B), Pielou_e (C), and Principal coordinate analysis (PCoA) (D) were assessed during the EFS process. Statistical assessment was carried out with one-way ANOVA followed by post hoc test using Dunn's multiple comparison test. Values with different letters are significantly different at $P < 0.05$.

showed higher degree, lower closeness centrality and higher average shortest path length (Fig 8C). Based on the networks, we identified keystones genus by MCC arithmetic in cytoHubba plugin in Cytoscape. *Unidentified_Methylophilaceae*, *Pseudoxanthomonas*, *Thermovum*, *unidentified_Sphingobacteriaceae*, *unidentified_Bacillales*, *Ureibacillus*, *Streptococcus*, *Megasphaera*, *unidentified_Weeksellaceae*, and *Syntrophomonas* were the key stone genus in bacteria network, while the 10 fungal genus turned to be keystone genus in fungal network (Fig 8D). Correlation analysis among keystone bacteria, fungi and environment factors was conducted. It showed moisture, C/N and TO were most relevant to bacteria and fungi (Fig 8D). Futhermore, *unidentified_Methylophilaceae*, *Ureibacillus* were most relevant to fungi and environment factors, and *Rhizormucor*, *Thermomyces* and *Aspergillus* were most relevant to bacteria and environment factors (Fig 8D). These findings showed that keystone bacteria and fungus interacted frequently, and moisture, C/N, and TO may have been the main environmental factors affecting bacterial and fungal alterations during EFS.

## Relationship between the bacterial and fungal community with environmental factors

Redundancy analysis was used to show the relationship between environmental factors (temperature, pH, TO, TN, C/N, moisture content and NN) for bacteria (Fig 9A) and fungus (Fig 9B), respectively. In the images, arrows denote various environmental factors, and the length of the rays demonstrates the impact size. The first axis accounted 41.61% and 55.68% of the variation in diversity, whereas the second axis explained 11.9% and 5.1% of the variation. The RDA biplot encompassed all seven of the environmental variables that were under investigation, indicating that they had a considerable impact on the EFS procedure. In the graphs, a positive connection is indicated by angles between the environment variables and sample dots that are smaller than 90 degrees. Smaller angles also showed stronger connections. Moisture

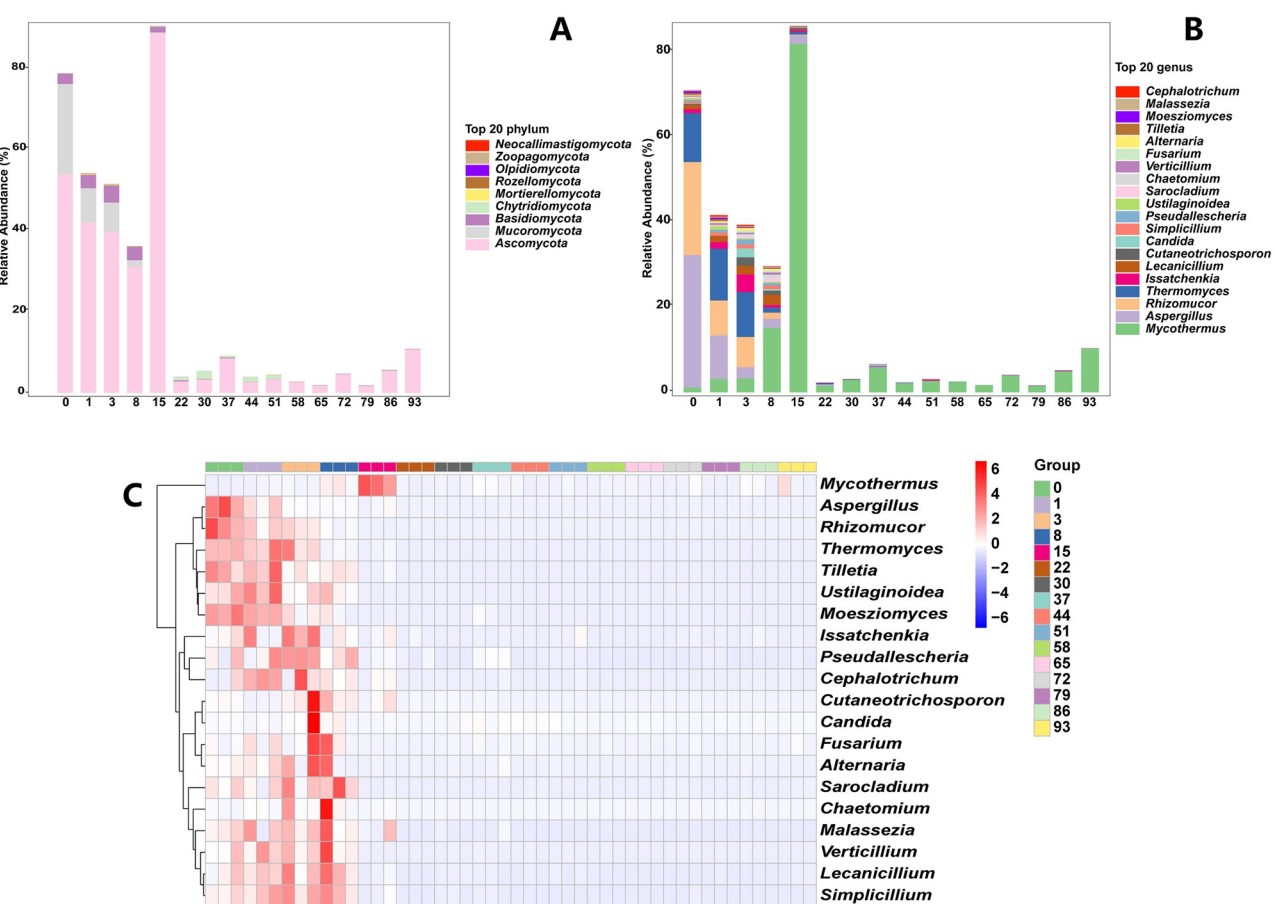

**Fig 7. Changes in fungal phylum and genus level and heatmap of top 20 genus in sample in ectopic fermentation of dairy manure.**

content, NN, and TN had a substantial impact on bacterial and fungal community change during the first 8 days, besides T and C/N had a significant impact from days 44 to 93.

## Discussion

In this study, we demonstrated changes of bacterial and fungal communities throughout the ectopic fermentation of dairy manure, and their correlations with the physicochemical variables. In our research, physicochemical factors not only control the rates of many biological processes but also have a selective role in the development and evolution of microbiological communities [9]. Nitrogen and organic carbon are essential for microbial growth and energy production during manure decomposition [26, 27], and the change of C/N ratio would effect on microorganism growth [28]. Additionally, the C/N ratio is regarded as an excellent indicator of the decomposition and fermentation, which the decrease means a higher level of decomposition and fermentation maturity [15, 29]. Thus, the results showed our fermentation achieved a high maturity level of decomposition and fermentation. During EFS process, conversion of urea to NN would raise the pH of fermentation mixes, but the acids generated by microorganisms will drop the pH through composting [30]. Besides, the pH level would be affected by the moisture change which caused by water evaporation [28]. Our results showed that pH change was relatively consistent with NN, suggesting that microbial denitrification

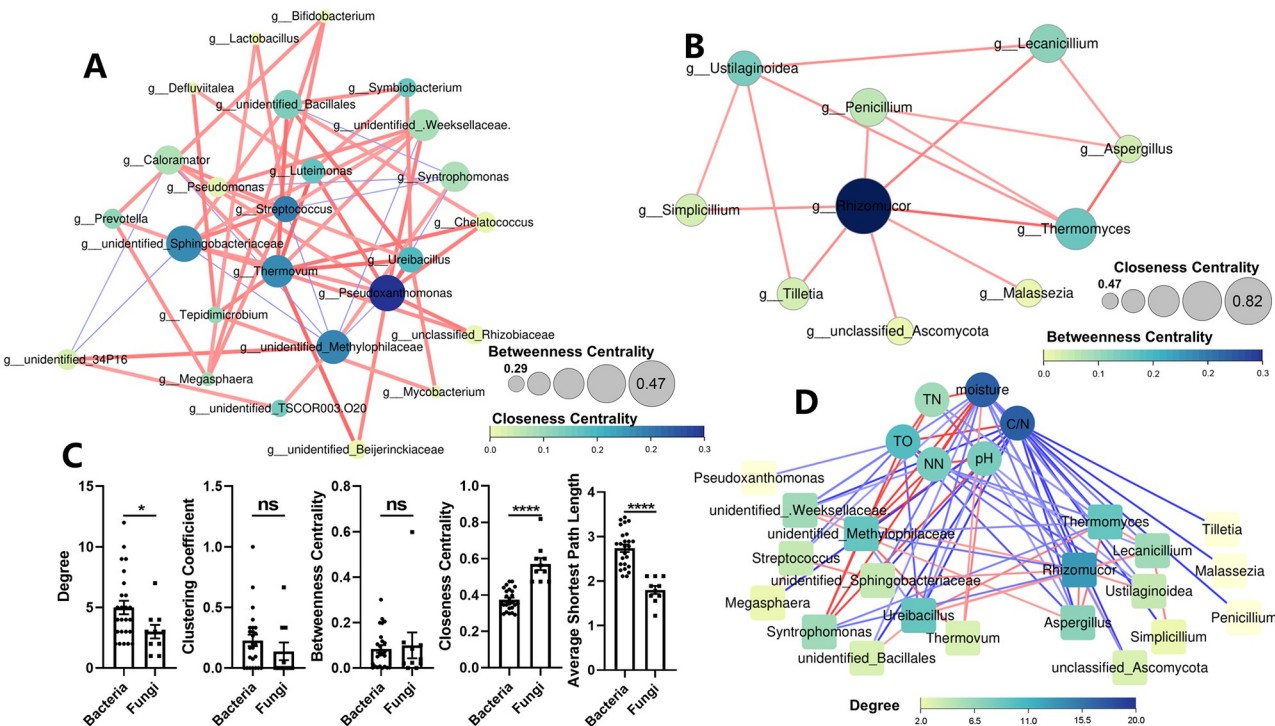

**Fig 8. Network analysis of bacterial and fungal community and correlation between keystone genus and environmental factors.** In bacterial (A) and fungal (B) network, nodes in the network represent taxon (genus level), and node size is proportional to closeness centrality, while node color is proportional to betweenness centrality; blue edge indicates negative correlation and red edge indicates positive correlation. Topological properties of nodes in bacterial and fungal networks (C). Correlation among keystone genus in bacterial and fungal network and environmental factors (D), and blue edge indicates negative correlation and red edge indicates positive correlation, while node color is proportional to degree.

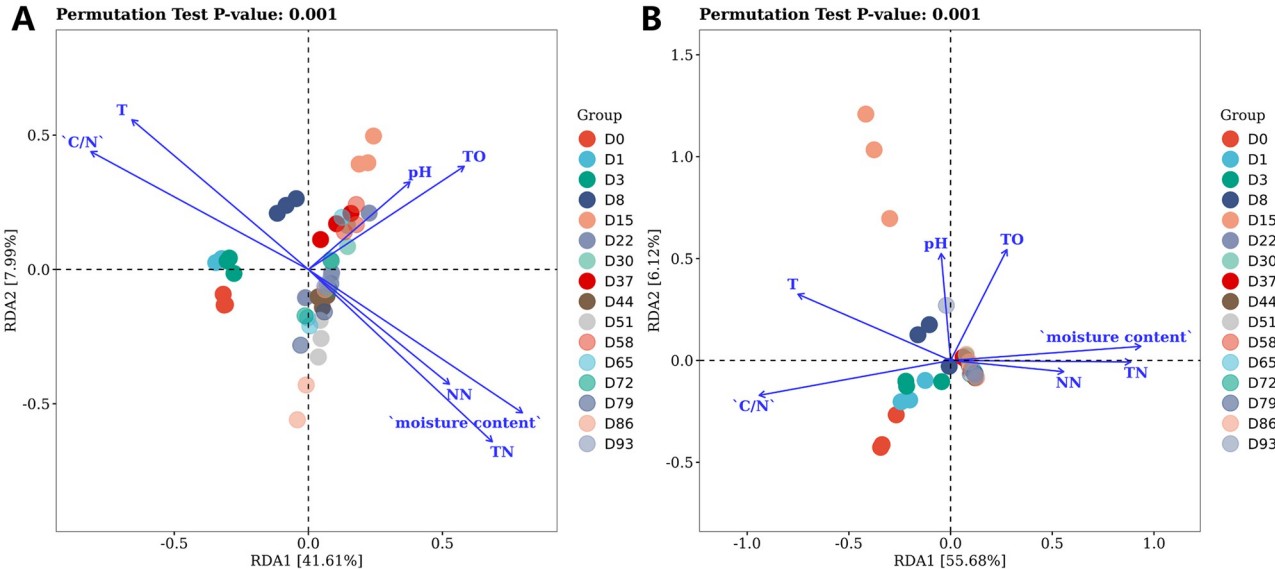

**Fig 9. Redundancy analysis (RDA) of bacterial communities and environmental factors.** RDA used to assess the relationships of bacterial communities (A) and environmental factors, as well as fungal communities (B) and environmental factors during the EFS process. The values of axes are the percentages explained by the corresponding axis.

may be more crucial for pH change. In addition, pH fluctuations might affect the development of the microorganisms and fermentation process [31]. Overall, the data indicated that our fermentation process was mature and at a high degree of decomposition based on the changes in physicochemical properties throughout fermentation.

Enzymatic activity was another indicator for the biological processes and will help your explanation of mechanism underlying organic matter degradation. Cellulases is involved in the process of C mineralization and is in charge of cellulose degradation, which is influenced by cellulolytic bacteria [32]. An earlier study found that total aerobic bacteria, aerobic cellulolytic fungus, and total aerobic celulllolytic bacteria were all positively linked with cellulases enzymatic activity [33]. The peak value of cellulases enzymatic activity could attribute to increased temperature and activity of thermophilic bacteria, while the enzymatic activity maintained at 18 U/g during later period indicated the end of fermentation. Urease is involved in the hydrolysis of urea to ammonium and carbon dioxide [34] and related to nitrogen cycle [35]. According to reports, urease and cellulases collaborated and were essential to the EFS [33, 36]. Dehydrogenase is crucial for all microbiota since it participates in the respiratory chain and is frequently used as a measure of total microbial activity [34]. The identical variable tendency of cellulases and dehydrogenase suggested microorganism activities of organic matter degradation and the C consumption. Since alkaline phosphatase is only produced by microbes and not by plants, it is very important for evaluating the composting process [37]. These findings suggested that microbial bioactivity would be crucial for the organic materials breakdown of fermentation.

Alpha diversity index (Chao1, Shannon and Pielou_e index) described the bacterial species richness, diversity and evenness, and reflected dynamic changes of microbial community during nitrogen transformation and surrounding changes during dairy manure ectopic fermentation [38]. The main trend of Chao1 index was increased, which was opposite to C/N ratio. These findings showed that greater species richness would promote fermentation maturity. During EFS, the declining of community diversity and evenness of denitrifying bacteria would lead to more nitrogen retention under the form of nitrate, which avoided excessive nitrogen release in the forms of NO, $N_2O$, and $N_2$. In these gases, NO is toxic and $N_2O$ is a major greenhouse gas, with an approximately 300-fold higher potential for global warming than $CO_2$ of equal mass [39]. Although our results showed the overall community diversity changes, they also indicated the effect of reducing environment pollution. After fermentation, we found the main pathogenic microorganisms showed an overall downward trend with slightly fluctuating, and the fecal coliform was increased significantly. Thus, the risk of pathogenic microorganisms and fecal coliform entering environment from dairy manure would be reduced after ectopic fermentation.

Due to changes of ambient variables and physicochemical properties during the fermentation, the original dominating microbial taxa associated with the compost materials eventually underwent alterations. In our study, *Firmicutes* and *Proteobacteria* were the most dominant phylum at first 8 days, and *Firmicutes* was slightly increased. Previous study showed that *Firmicutes* was dominant and increased at early stage, which could adapt to different composting systems utilizing various organic wastes [40]. *Proterobacteria* was dominant during the whole fermentation, and it was most abundant at early stage followed by a decrease and then increasing. Many classes or orders in *Proteobacteria* could be enriched in various stage of fermentation, which suggests that certain mesophilic bacteria were dormant during the thermophilic phase and came back to life during the cooling phase [41]. Also, *Proteobacteria* could degrade high-molecular-weight polysaccharides such as starch and cellulose. The *Chloroflexi* increased gradually and became the most dominant from day 15. *Chloroflexi* could contribute major

function genes of nitrate reductase, which plays key role in denitrification processes of reduction $NO_3^-$-N to $NO_2^-$-N and was positively correlated with NN [28].

In terms of fungus, the most dominant phylum was *Ascomycetes* overall the composting process, while *Mycothermus*, *Aspergillus*, *Rhizormucor*, *Thermomyces* were enriched at first 3 days. Besides, *Mycothermus* eventually became the most dominant taxa. *Thermomyces* was related to hemicellulose degradation [42, 43]. In single-stage inoculation, *Aspergillus* was the most prevalent fungus, demonstrating the significance of lignocellulose degradation [42, 44]. Moreover, on the composting of herbal residues, *Aspergillus* served as the primary fungus throughout the second high-temperature composting phase, which released cellulases and hemicellulases to break down the remaining lignocellulose after the thermophilic stage [45, 46]. Besides, previous studies also indicated that *Aspergillus* and *Mycothermus* were also frequently found in composting fermentation [44, 47–49]. Additionally, *Mycothermus* was dominant throughout the thermophilic stage of dairy manure composting and released hydrolases to take part in destruction of cellulose and hemicellulose [50]. In addition, *Mycothermus* was also dominant during thermophilic stage of swine manure and rice straw co-composting [51], which this thermophilic fungus commonly found in composting secreting heat-resistant enzymes to degrade lignocellulose [52]. Together, the fungus community performed a crucial and essential role in the degradation of different organic materials, and identified potential species for efficiently enhancing the composting process.

Network research revealed that the bacterial network had a higher average degree, which indicated that it was more intricate and closely connected with other genera. Closeness centrality represents a node that is closed to all other nodes in the network and indicates the information transmission between nodes in a high speed and efficient [53], while the shortest path indicates how quickly information can travel between two nodes [54, 55]. Compared to bacteria, fungi may react to environmental changes more quickly. Both the bacterial and fungal networks exhibited a predominance of positive correlations, suggesting that mutual cooperation played a more significant role in microbial formation during EFS. It was indicated that bacterial communities regularly engaged in competition with one another during EFS, because the bacterial network had more negative linkages than the fungal network [56]. The co-occurrence network showed the glimpse of community relationships, which demonstrated that bacteria were more closely related to one another than fungus during dairy manure ectopic fermentation. This suggested that bacterial communities kept a more stable status based on corporation and competition, which also disclosed the reason of lower fungi community diversity. The keystone bacterial and fungal genus was closely related to physiochemical parameters suggested that the fermentation-induced changes in microorganisms were what drove the composting process.

Also, there are still some limitation of this study. Firstly, we did not set a control or treatment to examine how physiochemical qualities or environmental conditions affected the composting process. Secondly, the analysis of bacteria and fungi were based on 16S rRNA and 18S rRNA gene sequence, which were inadequate to disclose accurate species, strains, and microbial functions. We did not culture the potential key species and verified their function in improving composting process. Despite the shortcomings, our work can be favorable to understand dynamic changes of bacterial and fungal community in composting process. Besides, some of the key species for decomposition of organic materials or the nitrogen cycle might be used as possible intervention methods to enhance the composting process.

## Conclusion

Our findings indicated that during the ectopic fermentation, the major pathogenic bacteria reduced while fecal coliform increased. The maturation and stability of fermentation were

indicated by changes in the physicochemical parameters and enzymatic activity, particularly the C/N ratio, cellulase, and dehydrogenase. Based on 16S and 18S rRNA gene sequencing, we discovered that bacterial phyla *Chloroflexi*, *Firmicutes* and *Proteobacteria* and fungal phylum *Ascomycota* were predominant in compost samples from EFS. Additionally, the diversity of bacteria increased while the diversity of fungus decreased. Furthermore, we found keystone bacterial and fungal taxon, such as *Unidentified_Methylophilaceae*, *unidentified_Bacillales*, *Aspergillus*, *Thermomyces* and *Rhizomucor*, were closely associated to physicochemical factors. Hence, this research indicated potential usage of keystone taxa and management of physico-chemical factors could contribute to maturation rate and quality improvement during dairy manure ectopic fermentation.

## Supporting information

**S1 Data.**
(XLSX)

## Acknowledgments

We thank Institute of Animal Husbandry and Veterinary, Wuhan Academy of Agricultural Science and Shanghai Personal Biotechnology Co., Ltd. (China) for their assistance.

## Author Contributions

**Conceptualization:** Ping Gong, Xiuzhong Hu.

**Formal analysis:** Ping Gong, Daoyu Gao, Junjun Tan.

**Funding acquisition:** Erguang Jin.

**Investigation:** Xiuzhong Hu, Junjun Tan, Lijun Wu.

**Methodology:** Junjun Tan, Yu Yang, Erguang Jin.

**Resources:** Wu Liu, Erguang Jin.

**Software:** Lijun Wu.

**Supervision:** Wu Liu.

**Writing – original draft:** Ping Gong.

**Writing – review & editing:** Ping Gong, Yu Yang.

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
