## [Decision Letter · Decision Letter 0]

6 Jun 2022

PONE-D-22-13572Changes of bacterial and fungal communities and relationship between keystone taxon and physicochemical factors during dairy manure ectopic fermentationPLOS ONE

Dear Dr. Jin,

Thank you for submitting your manuscript to PLOS ONE. After careful consideration, we feel that it has merit but does not fully meet PLOS ONE’s publication criteria as it currently stands. Therefore, we invite you to submit a revised version of the manuscript that addresses the points raised during the review process.

We look forward to receiving your revised manuscript.

Kind regards,

Sartaj Ahmad Bhat, Ph.D

Academic Editor

PLOS ONE

Journal Requirements:

Note that it is not acceptable for the authors to be the sole named individuals responsible for ensuring data access.

6. Please remove your figures from within your manuscript file, leaving only the individual TIFF/EPS image files, uploaded separately.  These will be automatically included in the reviewers’ PDF.

Additional Editor Comments:

 The manuscript needs considerable revisions according to the reviewer comments. Authors are encouraged to cite latest references in the manuscript.

Reviewers' comments:

Reviewer's Responses to Questions

**Comments to the Author**

1. Is the manuscript technically sound, and do the data support the conclusions?

Reviewer #1: Yes

Reviewer #2: Yes

2. Has the statistical analysis been performed appropriately and rigorously? 

Reviewer #1: Yes

Reviewer #2: Yes

3. Have the authors made all data underlying the findings in their manuscript fully available?

Reviewer #1: Yes

Reviewer #2: Yes

4. Is the manuscript presented in an intelligible fashion and written in standard English?

Reviewer #1: Yes

Reviewer #2: Yes

5. Review Comments to the Author

Reviewer #1: Dear Authors,

Your manuscript has been written well but it needs some corrections and improvements following the comments on the attached file. beside that, please read the text several times by yourself and by an expert, becouse there are frequently miss sentences and phrases.

Reviewer #2: The present study depicts the Changes of bacterial and fungal communities and the relationship between keystone taxon and physicochemical factors during dairy manure ectopic fermentation.

Abstract: should be divided into three sections: background, material methods, results discussion and conclusion. Please rewrite it.

Page 7 ligne 21: why the C/N ratio is negatively correlated with microorganisms ?

Page 11 ligne 81-83: remove space

Page 15 ligne 174: kindly remove "D";

page 15 ligne 179: please write the whole word (day). please review all the text;

page 17 ligne 214: kindly explain more about Pielou_e index which

page 23 ligne 307: environment factors instead of enrironment factors.

Please add the enzymatics activities identification in your manuscript that will help you to explain more the mechanism of organic matter degradation:

kindly check this reference you can added in your manuscript:

Saloua Biyada, Mohammed Merzouki, Taisija Dėmčėnko, Dovilė Vasiliauskienė, Eglė Marčiulaitienė, Saulius Vasarevičius, Jaunius Urbonavičius. The effect of feedstock concentration on the microbial community dynamics during textile waste composting. Frontiers in ecology and evolution (2022). 10, 813488, https://doi.org/10.3389/fevo.2022.813488.

Saloua Biyada, Mohammed Merzouki, Taisija Dėmčėnko, Dovilė Vasiliauskienė, Rūta Ivanec-Goranina, Jaunius Urbonavičius, Eglė Marčiulaitienė, Saulius Vasarevičius, Mohamed Benlemlih. Microbial community dynamics in the mesophilic and thermophilic phases of textile waste composting identified through next-generation sequencing. Scientific Reports (2021). 11, 23624. https://doi.org/10.1038/s41598-021-03191-1.

DISCUSSION:What is exactly the value add of this manuscript, a lot of study depicts the microbial communities in compost mixture.

The conclusion needs to be further improved.

English needs to be further improved.

6. PLOS authors have the option to publish the peer review history of their article (what does this mean?). If published, this will include your full peer review and any attached files.

Reviewer #1: No

Reviewer #2: No

---

## [Author Response · Author response to Decision Letter 0]

29 Aug 2022

Line 10: diversity of ?

Response: Thanks for your advice and we are sorry for the mistake. We have revised it to “composition and diversity of microbiota”, and please see line 12.

Line 16: phyum？

Response: Thanks for your advice and we feel sorry for the spelling mistake. We have revised it to “Phyla”. Please see line 28.

Line 26: “Our research” revised to “These findings”

Response: Thanks for the comments. We have revised it, and please see line 37.

Line 34: “the cattle stock achieved about 0.09 billion in 2018” not understandable

Response: Thanks for the comments and we are sorry for the writing mistake leading to misunderstanding the meaning. We have revised this part based on our re-searched reference. Please see Line 42-43.

Line 40-41: “which would also a huge threaten to environment and human health” rewrite better

Response: Thanks for the constructive advice. We have discussed this part and revised it. Please see Line 49-50.

Line 42-43: “methods to deal with livestock manure pollution would be needed to handle the waste and the urgent concerns caused by livestock waste”建议改成 management of livestock manure is quite urgent to facilitate its handling, transportation and application with limited environmental pollution (Souri et al., 2018; Souri et al., 2019).

Response: We appreciate for the constructive advice and instruction. We have re-read our work and re-wrote these part, and referred the proposed references. Please see Line 51-52 and reference 5-6. 

Souri, M.K., Naiji, M. and Kianmehr, M.H., 2019. Nitrogen release dynamics of a slow release urea pellet and its effect on growth, yield, and nutrient uptake of sweet basil (Ocimum basilicum L.). Journal of plant nutrition, 42(6), pp.604-614.

Souri, M.K., Rashidi, M. and Kianmehr, M.H., (2018). Effects of manure-based urea pellets on growth, yield, and nitrate content in coriander, garden cress, and parsley plants. Journal of Plant Nutrition, 41(11):1405-1413.

Line 46-51: R1：”Biological fermentation bed with chaff, pieces of straw, sawdust, mushroom residue, rice chaff, a moderate amount of microorganism strains as raw material, mixing after laying in the inner packing bed, cow excretion of feces to be absorbed by the padding and through microbial rapid decomposition, so as to achieve the aim of no manure emissions, after use, packing can be as raw materials for the production of organic fertilizer or soil use” 

not understandable; rewrite better

Response: We appreciate for the constructive advice and instruction. We have re-read our work and re-organized this part as follows: 

Ectopic fermentation system (EFS), composed of complex microbial communities mixed with litter, was developed to overcome the problem of pollution from existing farm wastewater systems [1], which has been proven by several studies in effectively managing pollution problems about livestock wastes such as cow, pig and so on [2-5]. The EFS is an experimental apparatus padded with bulking materials such as straw, sawdust, and rice husk, which is dynamically fermented by inoculation with a composite microbial agent and the continuous ad dition of livestock wastes [6]. As an ex situ decomposition technology, EFS has gained increasing concern from applied researchers and engineers with the characteristics of large-scale disposal of organic solid wastes [5]. 

Please see Line 53-61. 

Line 67: “litter research focus on bacteria and fungi interaction nor a long time dynamic change” rewrite better

Response: We appreciate for the constructive advice. We have re-considered our work and revised this part as follow: rare study revealed dynamic changes and network properties about bacteria and fungi, also the relation among bacteria, fungi and environmental and physicochemical parameters during EFS. Please see Line 76-78.

Line 76: “EFSs” ?!! 

Response: Thank you for the advice, we have revised it to “This”. Please see Line 87.

Line 78-79 “The main raw material was dairy mature and sewage mixture for ectopic fermentation” i can not understand!! rewrite better

Response: Thank you for the advice, which is critical and very hopeful to improve the quality of this manuscripts. We have re-read and re-checked carefully, and revised this part for a better understanding of our fermentation protocol. The revised words are as follow: “To create the EFS bed, rice husk and rice chaff was mixed at a ratio 6:4 (v/v) as mattress materials. Dairy waste mixed by manure and sewage was periodically added onto the bedding material and turned by the upender for fully mixing, loosening and ventilation to provide oxygen for pig manure fermentation and promote the degradation.” Please see Line 88-92.

Line 110: intestinal contents, what you mean with this? feaces?

Response: Thank you for the comment and we are sorry for the mistake. we have revised it this part. Please see Line 128.

Line 162: cow manure; to be similar write or dairy or cow manure but not both

Response: Thank you for the advice, we have re-read our manuscript carefully and revised “cow manure” to “dairy manure” in the text. Please see Line 180, 199, 225, 238, 259, 269, 326, 368, 380, 408, 428 and 457.

Line 173: “And, from” revised to “From” 

Response: Thank you for the advice. We have revised it and please see Line 185.

Line 347: Nitrogen and organic carbon are essential for microbial growth and energy production. Kindly recommended the references “during manure decomposition (Naiji and Souri, 2018; Serri et al., 2021)”,

Serri, F., Souri, M.K. and Rezapanah, M., 2021. Growth, biochemical quality and antioxidant capacity of coriander leaves under organic and inorganic fertilization programs. Chemical and Biological Technologies in Agriculture, 8(1), pp.1-8.

Naiji, M. and Souri, M.K., (2018). Nutritional value and mineral concentrations of sweet basil under organic compared to chemical fertilization, J of Hortorum Cultus, 17(2): 167175.

Response: We appreciate for the constructive advice and instruction. We have re-read our work and revised this part, while referred the proposed references. Please see line 331 and reference 26-27. 

Line 355-356 i think it can influence EC but not pH

Response: Thank you for the comment and the constructive advice. As far as I know, changes of electrical conductivity (EC) could attribute to volatilization of ammonia and the precipitation of minerals caused by the decomposition of OM, and related molecules included NH4+, NO3-, inorganic cations, H+, OH- and organic acids which affected by microbial metabolism [7]. Based on the reviewer’s reasonable comment, moisture change would have a profound effect on the concentration mentioned above to change the EC which was consisted with previous report [8]. In addition, moisture change during fermentation and microbial metabolism could also affect concentration of organic acids to influence pH which was also support by Chen et al 2021, and previous researches have reported opposite changes of pH and EC during composting [7, 9]. Taken together, moisture, pH and EC was interplayed, while we neglected to explore the EC changes during fermentation, which would be a defect of our experiment design. 

Line 385-386 “dominant at not only early stage, but also during the whole fermentation”, rewrite better 

Response: Thank you for the advice. We have re-check our results about changes of Proterobacteria, and revised the summary of the change. Please see Line 387-388.

Reviewer #2: The present study depicts the Changes of bacterial and fungal communities and the relationship between keystone taxon and physicochemical factors during dairy manure ectopic fermentation.

Abstract: should be divided into three sections: background, material methods, results discussion and conclusion. Please rewrite it.

Response: Thank you for the advice, which is critical and very hopeful to improve the quality of this manuscripts. We have re-read and re-checked carefully, and revised this part for a better understanding of our work. Please see section abstract, Line 12-38. 

Page 7 ligne 21: why the C/N ratio is negatively correlated with microorganisms ?

Response: Thank you for the comment. The C/N ratio is considered a good indicator of the quality of decomposition [10] and used to assess maturity and stability of fermentation [11]. In some researches, C/N ratio decreased at the end and/or at the later period of the experiment and fermentation, meanwhile organic matter content decreased indicating the degradation of organic substrates [1, 4, 8, 12, 13] (Ref A-E). During the fermentation, available carbon and volatilization of ammonia, which would affect C/N ration, were influenced by temperature and moisture, indicating an interaction between C/N and temperature, moisture [12]. The temperature and moisture would be affected be the activity of microorganisms during fermentation, such as Thermophilic bacteria activity raises the temperature and lead to water evaporate. On the other side, the keystone bacteria and fungi in the community could metabolize and degrade organic substrates. For example, Unclassified_Methylophilaceae could utilize single-carbon methyl substrates for denitrification [14], and the predominant Unclassified_Methylophilaceae resulted a low C/N ratio [15]. Bacteria belonging to the orders Bacillales possess the capacity to metabolize more recalcitrant substrates, such as cellulose and lignin, resulted to rapidly degradable organic compounds in the raw materials leading to decreased C/N ratio [16]. In addition, fungi are primarily involved in the decomposition of cellulose, hemicellulose and lignin present in the organic matter [17], which would lead to decreased organic matter resulted in lower C/N (Ref E). Saccharomonospora, Aspergillus, Thermomyces and Microascus were played a key role in promoting composting maturity [18]. This indicated that fungi such as Aspergillus and Thermomyces would also played key role in lower C/N ratio. Taken together, microorganism played vital role in degrading organic matter and affecting physiochemical condition based on their metabolic function and activity to decrease C/N ratio, while the keystone taxa showed negatively correlated with C/N ratio indicated their potential key role in promoting EFS maturity and stability.

Page 11 ligne 81-83: remove space

Response: Thank you for the advice and we are sorry for the mistake. we have revised it this part. Please see Line 93-95.

Page 15 ligne 174: kindly remove "D";

Response: Thanks for the advice and we are sorry for the mistake. we have revised it this part. Please see Line 186.

page 15 ligne 179: please write the whole word (day). please review all the text;

Response: We appreciate for the constructive advice and we have re-check our manuscripts. We have revised relative parts in the text. Please see Line 190-191 and 196-198.

page 17 ligne 214: kindly explain more about Pielou_e index which

Response: Thank you for the constructive advice. We have re-consider the biological meaning and calculating method of Pielou_e evenness index. The Pielou Evenness Index is a common species evenness index based on the evenness of the distribution of importance between species [19]. Pielou_e evenness index calculated as: 

Pielou_e evenness index = Shannon index / lnS

Where S in the number of species [19]. Therefore, we have added and revised the simple explanation in Line 231-232. Hope it works and be satisfactory.

page 23 ligne 307: environment factors instead of enrironment factors.

Response: Thanks for the advice and we are sorry for the mistake. we have revised it this part. Please see Line 305.

Please add the enzymatic activities identification in your manuscript that will help you to explain more the mechanism of organic matter degradation:

Response: Thank you for the advice, which is critical and very hopeful to improve the quality of this manuscripts. We have added the enzymatic activities identification of cellulose, urease, dehydrogenase and alkaline phosphatase. Cellulases is related to C mineralization and responsible for degrading cellulose, a process that depends on the types of cellulolytic microorganisms [20]. Urease is involved in the hydrolysis of urea to ammonium and carbon dioxide [21] and related to nitrogen cycle [22]. Dehydrogenase, which is used as a measure of overall microbial activity, is important for all microorganisms because it is involved in the respiratory chain [21]. Alkaline phosphatase is particularly relevant for the evaluation of the composting process, since it is only synthesized by microorganisms and does not originate from plant residues [23].

The results of cellulose, urease, dehydrogenase and alkaline phosphatase was shown in Figure 1. Cellulase activity gradually increased at first 15 days and decreased in day 22, while a slightly increase from day 30 to day 44 (Fig 1.A). During the fermentation, urease it fluctuated and showed three obvious peak value at day 1, day 22 and day 51, respectively (Fig 1.B). Dehydrogenase activity was rapidly increased to the peak value from day 0 to day 18 and dropped in the next 14 days, but it decreased from day 44 to day 93 (Fig 1.C). Alkaline phosphatase activity gradually decreased from day 0 to day 22, and increased in the next 43 days followed by a reducing (Fig 1.D). Please see line 214-224 and 346-364.

kindly check this reference you can added in your manuscript:

Saloua Biyada, Mohammed Merzouki, Taisija Dėmčėnko, Dovilė Vasiliauskienė, Eglė Marčiulaitienė, Saulius Vasarevičius, Jaunius Urbonavičius. The effect of feedstock concentration on the microbial community dynamics during textile waste composting. Frontiers in ecology and evolution (2022). 10, 813488, https://doi.org/10.3389/fevo.2022.813488.

Saloua Biyada, Mohammed Merzouki, Taisija Dėmčėnko, Dovilė Vasiliauskienė, Rūta Ivanec-Goranina, Jaunius Urbonavičius, Eglė Marčiulaitienė, Saulius Vasarevičius, Mohamed Benlemlih. Microbial community dynamics in the mesophilic and thermophilic phases of textile waste composting identified through next-generation sequencing. Scientific Reports (2021). 11, 23624. https://doi.org/10.1038/s41598-021-03191-1.

Response: We appreciate for the constructive advice and instruction. We have re-read our work and the proposed references. We thought the two articles were outstanding and we referred the proposed references. Please see reference 29 and 46. 

DISCUSSION:What is exactly the value add of this manuscript, a lot of study depicts the microbial communities in compost mixture.

Response: Thank you for the comment, which is critical and very helpful to improve the quality of this manuscripts. We have re-read our work and re-considered our discussion, and we would like to try our best to answer this question and hope it works.

Based on literature research during our study design and manuscripts preparing, we indeed found lots of work depicts the microbial communities during compost mixture, including bacteria and fungi. Nevertheless, rare study has paid attention to the network properties and keystone species of microbial communities, and other similar researches commonly showed changes of diversity, composition and biomarker taxa at different stages. While the compost or ectopic fermentation process was an ideal dynamic and longitudinal study model for microbial ecology and evolution, we aim to reveal the community network properties and keystone species based on SparCC algorithm, which is a typical method for network analysis. Recent studies showed keystone taxa are highly connected taxa that individually or in a guild exert a considerable influence on microbiome structure and functioning irrespective of their abundance across space and time, which have a unique and crucial role in microbial communities [24, 25]. Under this circumstance, this manuscript revealed keystone bacteria and fungi which could play vital role in promoting maturity and stability, degrading substrates, and detoxing hazardous substance in the compost mixture during the compost or ectopic fermentation process, although there was still some limitation of this study. Further research is warrant to explore the mechanism and function of the keystone species during compost or ectopic fermentation process. 

The conclusion needs to be further improved.

Response: Thanks for the constructive comment, which is critical, important and very meaningful to improve the quality of this manuscripts. We have re-read and re-considered our work, and we would like to revised it. Please see line 445-457. Hope it reasonably concluded and summarized our works.

English needs to be further improved.

Thank you for the comment, which is critical and very helpful to improve the quality of this manuscripts. We carefully re-read our manuscripts and asked for help to some people who had studied in the country speak English and English teacher who is expertise in in science and technology paper writing. Then, we have revised some language mistakes and sentences with unclear expression.

Reference

1. Guo H, Geng B, Liu X, Ye J, Zhao Y, Zhu C, et al. Characterization of bacterial consortium and its application in an ectopic fermentation system. Bioresource technology. 2013;139:28-33. Epub 2013/05/07. doi: 10.1016/j.biortech.2013.04.029. PubMed PMID: 23644067.

2. Shen Q, Sun H, Yao X, Wu Y, Wang X, Chen Y, et al. A comparative study of pig manure with different waste straws in an ectopic fermentation system with thermophilic bacteria during the aerobic process: Performance and microbial community dynamics. Bioresource technology. 2019;281:202-8. Epub 2019/03/02. doi: 10.1016/j.biortech.2019.01.029. PubMed PMID: 30822641.

3. Shen Q, Tang J, Wang X, Li Y, Yao X, Sun H, et al. Fate of antibiotic resistance genes and metal resistance genes during the thermophilic fermentation of solid and liquid swine manures in an ectopic fermentation system. Ecotoxicol Environ Saf. 2021;213:111981. Epub 2021/02/17. doi: 10.1016/j.ecoenv.2021.111981. PubMed PMID: 33592372.

4. Yang X, Geng B, Zhu C, Li H, He B, Guo H. Fermentation performance optimization in an ectopic fermentation system. Bioresource technology. 2018;260:329-37. Epub 2018/04/11. doi: 10.1016/j.biortech.2018.03.101. PubMed PMID: 29635213.

5. Wen P, Wang Y, Huang W, Wang W, Chen T, Yu Z. Linking Microbial Community Succession With Substance Transformation in a Thermophilic Ectopic Fermentation System. Front Microbiol. 2022;13:886161. Epub 2022/05/24. doi: 10.3389/fmicb.2022.886161. PubMed PMID: 35602041; PubMed Central PMCID: PMCPMC9116721.

6. Li J, Liu X, Zhu C, Luo L, Chen Z, Jin S, et al. Influences of human waste-based ectopic fermentation bed fillers on the soil properties and growth of Chinese pakchoi. Environ Sci Pollut Res Int. 2022. Epub 2022/05/18. doi: 10.1007/s11356-022-20636-w. PubMed PMID: 35579832.

7. Chen P, Zhang L, Li Y, Liang J. Insight to maturity during biogas residue from food waste composting in terms of multivariable interaction. Environ Sci Pollut Res Int. 2022. Epub 2022/05/24. doi: 10.1007/s11356-022-20616-0. PubMed PMID: 35604592.

8. Chen Q, Wang J, Zhang H, Shi H, Liu G, Che J, et al. Microbial community and function in nitrogen transformation of ectopic fermentation bed system for pig manure composting. Bioresource technology. 2021;319:124155. Epub 2020/10/10. doi: 10.1016/j.biortech.2020.124155. PubMed PMID: 33035862.

9. Liu C, Zhang X, Zhang W, Wang S, Fan Y, Xie J, et al. Mitigating gas emissions from poultry litter composting with waste vinegar residue. Sci Total Environ. 2022;842:156957. Epub 2022/06/28. doi: 10.1016/j.scitotenv.2022.156957. PubMed PMID: 35760166.

10. Gu W, Zhang F, Xu P, Tang S, Xie K, Huang X, et al. Effects of sulphur and Thiobacillus thioparus on cow manure aerobic composting. Bioresource technology. 2011;102(11):6529-35. Epub 2011/04/13. doi: 10.1016/j.biortech.2011.03.049. PubMed PMID: 21482106.

11. Jurado MM, Suárez-Estrella F, Vargas-García MC, López MJ, López-González JA, Moreno J. Evolution of enzymatic activities and carbon fractions throughout composting of plant waste. J Environ Manage. 2014;133:355-64. Epub 2014/01/15. doi: 10.1016/j.jenvman.2013.12.020. PubMed PMID: 24412984.

12. Huhe, Jiang C, Wu Y, Cheng Y. Bacterial and fungal communities and contribution of physicochemical factors during cattle farm waste composting. Microbiologyopen. 2017;6(6). Epub 2017/07/25. doi: 10.1002/mbo3.518. PubMed PMID: 28736905; PubMed Central PMCID: PMCPMC5727367.

13. Liu D, Zhang R, Wu H, Xu D, Tang Z, Yu G, et al. Changes in biochemical and microbiological parameters during the period of rapid composting of dairy manure with rice chaff. Bioresource technology. 2011;102(19):9040-9. Epub 2011/08/13. doi: 10.1016/j.biortech.2011.07.052. PubMed PMID: 21835612.

14. Kalyuhznaya MG, Martens-Habbena W, Wang T, Hackett M, Stolyar SM, Stahl DA, et al. Methylophilaceae link methanol oxidation to denitrification in freshwater lake sediment as suggested by stable isotope probing and pure culture analysis. Environ Microbiol Rep. 2009;1(5):385-92. Epub 2009/10/01. doi: 10.1111/j.1758-2229.2009.00046.x. PubMed PMID: 23765891.

15. Shen Q, Wei J, Jiang L, Zhang Q, Mao Y, Liu C, et al. Denitrification performance and characteristics of untreated corncob for enhanced nitrogen removal of municipal sewage with low C/N ratio. Environ Res. 2022;213:113673. Epub 2022/06/17. doi: 10.1016/j.envres.2022.113673. PubMed PMID: 35710021.

16. Estrada-Bonilla GA, Lopes CM, Durrer A, Alves PRL, Passaglia N, Cardoso E. Effect of phosphate-solubilizing bacteria on phosphorus dynamics and the bacterial community during composting of sugarcane industry waste. Syst Appl Microbiol. 2017;40(5):308-13. Epub 2017/06/25. doi: 10.1016/j.syapm.2017.05.003. PubMed PMID: 28645701.

17. Raut MP, Prince William SP, Bhattacharyya JK, Chakrabarti T, Devotta S. Microbial dynamics and enzyme activities during rapid composting of municipal solid waste - a compost maturity analysis perspective. Bioresource technology. 2008;99(14):6512-9. Epub 2007/12/25. doi: 10.1016/j.biortech.2007.11.030. PubMed PMID: 18155903.

18. Lu X, Yang Y, Hong C, Zhu W, Yao Y, Zhu F, et al. Optimization of vegetable waste composting and the exploration of microbial mechanisms related to fungal communities during composting. J Environ Manage. 2022;319:115694. Epub 2022/07/17. doi: 10.1016/j.jenvman.2022.115694. PubMed PMID: 35841778.

19. Tu HM, Fan MW, Ko JC. Different Habitat Types Affect Bird Richness and Evenness. Sci Rep. 2020;10(1):1221. Epub 2020/01/29. doi: 10.1038/s41598-020-58202-4. PubMed PMID: 31988439; PubMed Central PMCID: PMCPMC6985263.

20. Hemati A, Aliasgharzad N, Khakvar R, Khoshmanzar E, Asgari Lajayer B, van Hullebusch ED. Role of lignin and thermophilic lignocellulolytic bacteria in the evolution of humification indices and enzymatic activities during compost production. Waste Manag. 2021;119:122-34. Epub 2020/10/16. doi: 10.1016/j.wasman.2020.09.042. PubMed PMID: 33059162.

21. Castaldi P, Garau G, Melis P. Maturity assessment of compost from municipal solid waste through the study of enzyme activities and water-soluble fractions. Waste Manag. 2008;28(3):534-40. Epub 2007/03/27. doi: 10.1016/j.wasman.2007.02.002. PubMed PMID: 17382530.

22. Mondini C, Fornasier F, Sinicco T. Enzymatic activity as a parameter for the characterization of the composting process. Soil Biology and Biochemistry. 2004;36(10):1587-94.

23. Burns RG. Enzyme activity in soil: location and a possible role in microbial ecology. Soil biology and biochemistry. 1982;14(5):423-7.

24. Banerjee S, Walder F, Büchi L, Meyer M, Held AY, Gattinger A, et al. Agricultural intensification reduces microbial network complexity and the abundance of keystone taxa in roots. The ISME journal. 2019;13(7):1722-36. Epub 2019/03/10. doi: 10.1038/s41396-019-0383-2. PubMed PMID: 30850707; PubMed Central PMCID: PMCPMC6591126.

25. Banerjee S, Schlaeppi K, van der Heijden MGA. Keystone taxa as drivers of microbiome structure and functioning. Nat Rev Microbiol. 2018;16(9):567-76. Epub 2018/05/24. doi: 10.1038/s41579-018-0024-1. PubMed PMID: 29789680.

---

## [Decision Letter · Decision Letter 1]

20 Sep 2022

PONE-D-22-13572R1Changes of bacterial and fungal communities and relationship between keystone taxon and physicochemical factors during dairy manure ectopic fermentationPLOS ONE

Dear Dr. Jin,

Thank you for submitting your manuscript to PLOS ONE. After careful consideration, we feel that it has merit but does not fully meet PLOS ONE’s publication criteria as it currently stands. Therefore, we invite you to submit a revised version of the manuscript that addresses the points raised during the review process.

ACADEMIC EDITOR: There are several grammatical and punctuation related errors throughout the manuscript, so authors must check the manuscript language and also rephrase the long sentences throughout the manuscript.Please ensure that your decision is justified on PLOS ONE’s publication criteria and not, for example, on novelty or perceived impact.

We look forward to receiving your revised manuscript.

Kind regards,

Sartaj Ahmad Bhat, Ph.D

Academic Editor

PLOS ONE

Journal Requirements:

Reviewers' comments:

Reviewer's Responses to Questions

**Comments to the Author**

1. If the authors have adequately addressed your comments raised in a previous round of review and you feel that this manuscript is now acceptable for publication, you may indicate that here to bypass the “Comments to the Author” section, enter your conflict of interest statement in the “Confidential to Editor” section, and submit your "Accept" recommendation.

Reviewer #1: All comments have been addressed

Reviewer #2: All comments have been addressed

2. Is the manuscript technically sound, and do the data support the conclusions?

Reviewer #1: Yes

Reviewer #2: Yes

3. Has the statistical analysis been performed appropriately and rigorously? 

Reviewer #1: Yes

Reviewer #2: Yes

4. Have the authors made all data underlying the findings in their manuscript fully available?

Reviewer #1: Yes

Reviewer #2: Yes

5. Is the manuscript presented in an intelligible fashion and written in standard English?

Reviewer #1: Yes

Reviewer #2: Yes

6. Review Comments to the Author

Reviewer #1: Dear Authors,

The manuscript has been well revised with no further comments while reading it. Now it can be accepted for possible publication.

Reviewer #2: line 26: rise instead of rose, please correct it. please check all typographical errors.

please improve the quality of the figures, they are invisible.

7. PLOS authors have the option to publish the peer review history of their article (what does this mean?). If published, this will include your full peer review and any attached files.

Reviewer #1: No

Reviewer #2: **Yes: **BIYADA Saloua

---

## [Author Response · Author response to Decision Letter 1]

8 Oct 2022

Reviewer #1: Dear Authors,

The manuscript has been well revised with no further comments while reading it. Now it can be accepted for possible publication.

Response: Thanks for your constructive advice and instruction during the peer review work, which is crucial and vital for our improving the manuscripts quality. We are also very appreciated for your affirmation and acceptance comments. 

Reviewer #2: line 26: rise instead of rose, please correct it. please check all typographical errors.

Response: Thanks for the comments and we are sorry for the writing mistake. We have revised it, please see line 26.

We have re-read our work and carefully check the manuscripts, and we have revised same typographical errors. Please see Line 23, Line 61, Line 107, Line 110, Line 338, Line 343, Line 346, Line 362, Line 364, Line 405 and Line 442.

please improve the quality of the figures, they are invisible.

Response: Thanks for your constructive advice and we are sorry for mistake and negligence during plotting. We have re-plot invisible figures, please see figure files.

Response to Academic Editor questions:

ACADEMIC EDITOR: There are several grammatical and punctuation related errors throughout the manuscript, so authors must check the manuscript language and also rephrase the long sentences throughout the manuscript.

Response: Thanks for the comments and advice. We have carefully re-read and checked our manuscripts under the assistance of native English speaker and people studied in USA for years. We have rephrased some sentences and revised grammatical and punctuation related errors. Please see Line 14-16, Line 17-19, Line 34-37, Line 53-57, Line 214-215, Line 217-219, Line 221-223, Line 225-226, Line 230-234, Line 241-242, Line 243- 244, Line 249-253, Line 261-268, Line 273-275, Line 291-294, Line 300-307, Line 310-313, Line 319-321, Line 323-325, Line 330-332, Line 335-336, Line 359-360, Line 366-369, Line 374-379, Line 389-392, Line 398-401, Line 407-408 and Line 420-422. Hope it meet the criteria for publication. 

Response: Thanks for the comments and advice. We have read and studied the publication criteria, and we ensured that this work and manuscripts is justified on PLOS ONE’s publication criteria.

Journal Requirements:

Response: Thanks for the comments and advice. We have rechecked our references in relevant databases such as Pubmed, Web of Science and so on. None of our cited references have been retracted.

---

## [Decision Letter · Decision Letter 2]

17 Oct 2022

Changes of bacterial and fungal communities and relationship between keystone taxon and physicochemical factors during dairy manure ectopic fermentation

PONE-D-22-13572R2

Dear Dr. Jin,

We’re pleased to inform you that your manuscript has been judged scientifically suitable for publication and will be formally accepted for publication once it meets all outstanding technical requirements.

Kind regards,

Sartaj Ahmad Bhat, Ph.D

Academic Editor

PLOS ONE

Additional Editor Comments (optional):

Reviewers' comments:

Reviewer's Responses to Questions

**Comments to the Author**

1. If the authors have adequately addressed your comments raised in a previous round of review and you feel that this manuscript is now acceptable for publication, you may indicate that here to bypass the “Comments to the Author” section, enter your conflict of interest statement in the “Confidential to Editor” section, and submit your "Accept" recommendation.

Reviewer #2: All comments have been addressed

2. Is the manuscript technically sound, and do the data support the conclusions?

Reviewer #2: Yes

3. Has the statistical analysis been performed appropriately and rigorously? 

Reviewer #2: Yes

4. Have the authors made all data underlying the findings in their manuscript fully available?

Reviewer #2: Yes

5. Is the manuscript presented in an intelligible fashion and written in standard English?

Reviewer #2: Yes

6. Review Comments to the Author

Reviewer #2: The manuscript has been well revised no further comments. so i believe Now it is suitable for publication.

7. PLOS authors have the option to publish the peer review history of their article (what does this mean?). If published, this will include your full peer review and any attached files.

Reviewer #2: **Yes: **Saloua Biyada

---

## [Editor Report · Acceptance letter]

9 Dec 2022

PONE-D-22-13572R2 

Changes of bacterial and fungal communities and relationship between keystone taxon and physicochemical factors during dairy manure ectopic fermentation 

Dear Dr. Jin:

I'm pleased to inform you that your manuscript has been deemed suitable for publication in PLOS ONE. Congratulations! Your manuscript is now with our production department. 

Kind regards, 

on behalf of

Dr. Sartaj Ahmad Bhat 

Academic Editor

PLOS ONE